# The Role of Oxidative Stress in Skeletal Muscle Myogenesis and Muscle Disease

**DOI:** 10.3390/antiox11040755

**Published:** 2022-04-11

**Authors:** Di Lian, Ming-Ming Chen, Hanyu Wu, Shoulong Deng, Xiaoxiang Hu

**Affiliations:** 1State Key Laboratory of Agrobiotechnology, College of Biological Sciences, China Agricultural University, Beijing 100193, China; b20173020099@cau.edu.cn (D.L.); wuhanyu@cau.edu.cn (H.W.); 2Beijing Key Laboratory for Animal Genetic Improvement, College of Animal Science and Technology, China Agricultural University, Beijing 100193, China; chenmingming1937@cau.edu.cn; 3NHC Key Laboratory of Human Disease Comparative Medicine, Institute of Laboratory Animal Sciences, Chinese Academy of Medical Sciences and Comparative Medicine Center, Peking Union Medical College, Beijing 100021, China

**Keywords:** ROS/RNS, oxidative stress, myogenesis, muscle atrophy, antioxidant therapy

## Abstract

The contractile activity, high oxygen consumption and metabolic rate of skeletal muscle cause it to continuously produce moderate levels of oxidant species, such as reactive oxygen species (ROS) and reactive nitrogen species (RNS). Under normal physiological conditions, there is a dynamic balance between the production and elimination of ROS/RNS. However, when the oxidation products exceed the antioxidant defense capacity, the body enters a state of oxidative stress. Myogenesis is an important process to maintain muscle homeostasis and the physiological function of skeletal muscle. Accumulating evidence suggests that oxidative stress plays a key role in myogenesis and skeletal muscle physiology and pathology. In this review, we summarize the sources of reactive oxygen species in skeletal muscle and the causes of oxidative stress and analyze the key role of oxidative stress in myogenesis. Then, we discuss the relationship between oxidative stress and muscle homeostasis and physiopathology. This work systematically summarizes the role of oxidative stress in myogenesis and muscle diseases and provides targets for subsequent antioxidant therapy and repair of inflammatory damage in noninflammatory muscle diseases.

## 1. Introduction

Oxidative stress (OS), defined as disturbance in the pro/antioxidant balance, is harmful to cells because it involves the excessive generation of highly reactive oxygen species (ROS) and reactive nitrogen species (RNS). Under normal physiological conditions, cells maintain redox homeostasis by generating and eliminating ROS/RNS. OS plays a role in physiological adaptation and signal transduction [1], including the regulation of cell survival, but if the redox balance is disrupted, the resulting severe increase in ROS/RNS leads to adverse changes to cell components. This, in turn, results in inflammatory neutrophil infiltration, increased secretion of proteases, and the production of a large number of oxidation intermediates, which are considered important factors in aging and disease [2]. As described in this review, scientists have found that OS is closely related to disease (kidney, nerves, brain) and inflammation [3,4,5]. Moreover since the outbreak of the Corona virus Disease 2019 (COVID-19), an increasing number of studies have focused on OS and COVID-19, including the associated tissue damage, pathogenesis and therapeutics [6,7,8].

Striated skeletal muscle is the most abundant tissue in the human body, representing approximately 35–45% of its total mass [9]. Skeletal muscle is a heterogeneous tissue composed of muscle fibers, basement membrane muscle satellite cells, and nerves [10], which are essential for vital functions such as movement, postural support, breathing, and thermogenesis.

Skeletal muscles have two remarkable capacities: adaptive potential and exceptional regenerative capacity. Loss of muscle function and impaired muscle regeneration ability not only severely reduces quality of life, but can also affect the careers of professional athletes. In addition to inflammatory muscle disorders, inflammation is also found in noninflammatory muscle conditions, such as those after high-intensity exercise or varying degrees of muscle damage. Such inflammation, is mainly due to elevated levels of ROS.

Accumulating evidence has demonstrated the effects (positive and negative) of OS on skeletal muscle and revealed the associated pathogenic mechanisms. However, the role of ROS in skeletal muscle beyond their effects on classical pathways is incompletely understood. In this review, we discuss the effects of ROS on skeletal muscle, describe the pathogenic mechanisms in noninflammatory skeletal muscle diseases, and summarize the current most commonly used antioxidant therapies. This paper should provide a reference for researchers to continue to study OS in skeletal muscle and to establish new treatment modalities.

## 2. Oxidative Stress and Reactive Oxygen

### 2.1. Causes and Sources of ROS and RNS

The imbalance between ROS and RNS production and elimination will cause OS. so it is crucial to understand where these species come from. ROS are produced by living organisms as a result of normal cellular metabolism. RNS are substances produced by the interaction of NO with compounds including reactive oxygen species. Depending on their chemical properties, ROS and RNS can be separated into two groups: radical and nonradical compounds form the second one [11].The radical ROS species include superoxide (O_2_^−^) and hydroxyl radicals (HO), while the nonradical ones include hydrogen peroxide (H_2_O_2_). RNS include nitric oxide (NO) and peroxynitirite (ONOO^−^) (Table 1).

### 2.2. Major Sources of ROS in Skeletal Muscle

#### 2.2.1. Mitochondrial

In organisms, most ROS are produced by mitochondria. Mitochondria are highly dynamic organelles, playing a role in oxidative phosphorylation and many cellular processes in most cell types. Skeletal muscle cells are rich in mitochondria, which are metabolically active and particularly likely to produce ROS. ROS generated as by-products of mitochondrial oxidative phosphorylation, are particularly damaging to the genome of skeletal muscle because of their high oxygen consumption. Under physiological conditions, 90% of the oxygen taken in by the body is oxidatively phosphorylated in the mitochondria, with about 1–2% escaping from the mitochondrial respiratory chain from reactive oxygen species. Excessive exercise produce more ROS and RNS, which can lead to impaired muscle contraction and cause muscle damage [15]. The mitochondrial transport chain (ETC) is the main source of ATP in mammalian cells, and ETC can produce O_2_^−^ at different locations on both sides of the mitochondrial inner membrane [16]. Under physiological conditions, the complexes I and III of ECT are the major production sites of O_2_^−^ [17]. The coupled respiration on glutamate/malate or pyruvate/malate will activate the tricarboxylic acid (TCA) cycle enzymes 2-oxoglutarate dehydrogenase (OGDH) malate dehydrogenase (MDH), and pyruvate dehydrogenase (PDH), and maintains a low membrane potential as complex V is generating ATP [18]. As the electrons flow down the ETC, they eventually reach complex III and IV. Complex II is succinate, a substrate for the TCA cycle enzyme succinate dehydrogenase(SDH) [19]. Moreover, under conditions of mitochondrial hyperpolarization, electrons return through complex I to reduce NAD+ to NADH.

Contractile activity is mainly performed by muscle fibers and there are differences in oxidative capacity due to variation in mitochondrial contents among different types of muscle fiber (fast-glycolytic, fast oxidative/glycolytic, and slow fibers). The number of mitochondria in the muscle determines the oxidative capacity of the muscle fibers. Fast contractions rely on glycolysis, which occurs in the cytosol and permits rapid ATP generation, but are inefficient. Slow contractions produces ATP through mitochondrial oxidative phosphorylation, which are slower, but more efficient [20]. Fast II muscle fibers have unique properties that promote higher levels of ROS production than slow type muscle fibers [21]. Researchers also found that Tap63γ participates in the control of myoblast metabolism: knockdown of Tap63 expression was shown to cause mitochondrial respiration, manifesting as reduced spare respiratory capacity and a decrease in complex I and IV protein levels, resulting in higher myoblast proliferation rate [22].

#### 2.2.2. NADPH Oxidases

NADPH oxidases (NOX) were first identified in neutrophils and macrophages, which can produce large amounts of ROS during the inflammatory response and constitute the body’s first line of defense against pathogens. NOX is a complex of six subunits: gp91phox, p22phox, p47phox, p67phox, p40phox and Rac among which gp91phox is the functional subunit. Homologs of pg91phox have been found in different cell types: NOX1, NOX2, NOX3, NOX4, NOX5, DUOX1 and DUOX2. In skeletal muscle cells, NOX isoforms 2 and 4 are the main source of ROS during skeletal muscle contraction, which are located in the plasma membrane, transvers striatum and sarcoplasmic reticulum, acting to regulate calcium release [23].

### 2.3. The Role of ROS in Skeletal Muscle

ROS are not necessarily harmful to cells (Figure 1). Accumulating evidence have shown that the antioxidants can put ROS in a optimal concentrations to perform physiological signal in muscle. At appropriate concentrations, ROS and RNS can regulate intracellular signal transduction [1]. Moderate levels of free radicals keep the body adaptive responses. Low levels of oxidation in the muscles at rest ensure normal force production, and oxidants during exercise enhance the body’s ability to resist damage. Myogenic cells are containing antioxidant enzymes, such as heme oxygenase-1, catalase, glutathione peroxidase and superoxide dismutase. These enzymes can neutralize excessive ROS, and play an important role in the regeneration process: influencing post-injury inflammatory reaction, affecting differentiation by enhancing satellite cell viability and proliferative capacity [24]. at the transcriptional level, ROS can stimulates the expression of factors involved in redox regulation and mitochondrial dynamics. For example, mitochondrial ROS levels are regulated by induction of PGC-1α/β-dependent antioxidant defense mechanisms of the mitochondria [25], and PGC-1α/β is redox-sensitive and associated with MFN2 regulation [26]. In addition, AMPK stimulates PGC-1α-dependent mitochondrial biogenesis. However, when AMPK is activated, MFF and DRP1 are phosphorylated and can mediate energy deprivation due to mitochondrial fission [27]. It has been proved that quiescent skeletal muscle satellite cells have lower ROS levels, but contain more antioxidants which help get far from harmful effects of potential ROS [28,29]. The increase in antioxidant enzymes leads to a decrease in ROS content late in the muscle differentiation process [23]. Mechanistically, excessive ROS in myoblasts can elevate nuclear factor kappa B (NF-κB) and decrease the expression level of MyoD, thereby inhibiting myogenic differentiation [30,31]. In addition to ROS that can help muscle fibers adapt to contractile activity, ROS produced by muscle contraction can also activate redox-sensitive signaling pathways that help skeletal muscle adapt to aerobic exercise. For example, exposure to hydrogen peroxide has been shown to augment the expression of key antioxidant enzymes in myotubes, such as Cu,Zn-superoxide dismutase(CuZnSOD), glutathione peroxidase(GPx), Mn-superoxide dismutase(MnSOD) [32].

## 3. Muscle Regeneration

Myogenesis occurs during both postnatal growth and the regeneration of skeletal muscle after injury. During embryonic muscle formation, muscle fibers arise from the mesoderm and subsequently other fibers are generated along these templates [33]. Adult skeletal muscle, by means of self-renewal of skeletal muscle satellite cells, compensates for the turnover of terminally differentiated cells, maintaining tissue homeostasis and balance [34]. Myogenesis is a very orderly process that involves, the activation of satellite cells into committed myoblasts, the proliferation and differentiation of myoblasts, and cell fusion to form myotubes [35,36]. It is significant to understand that the myogenic regulatory factors (MRFs) MyoD, Myf5, Myf6, and Myogenin, the myocyte enhancer factors (MEFs) and the serum response factor (SRF) play an important role in myogenesis [37].

Muscle regeneration is a characteristic of skeletal muscle on which many studies have focused. Numerous factors can influence muscle regeneration, such as aging and disease. The regeneration process requires the migration of an undifferentiated progenitor cells to the site of injury to perform the function, which in mature skeletal muscle is achieved by muscle satellite cells [38].

After muscle injury, lots of signals stimulate muscle satellite cells which cause them migrate toward the site of injury and re-enter the cell cycle to undergo proliferation. The classical Wnt signaling pathway drives the differentiation of muscle satellite cells mainly through the ligand Wnt3a, and the nonclassical Wnt signaling pathway regulates the differentiation of muscle satellite cells and self-renewal [39,40,41]. At the early stage of regeneration, Wnt7a binds to Fzd7 and activates the Wnt/PCP pathway via Vangl2, which regulate the symmetric division of muscle satellite cells and migration [42]. Lots of signals can promote muscle satellite cell proliferation, for example, LIF can reduces caspase 3 by activating the MEK/ERK signaling pathway thus stimulate proliferation [43]; IL-6 promotes proliferation by activating the JAK/STAT1/STAT3 signaling pathway [44] and PI3K/Akt signaling pathway not only promote muscle protein synthesis, but also activates the proliferation of muscle satellite cells [45,46]. The ensuing regeneration can be separated into several states that are characterized by the expression of different MRFs [47]. Some important pathways involved in this content should also mentioned: Pax7 is one critical postnatal regulator as its depletion results in the progressive loss of satellite cells during homeostasis and subsequent injury. The Notch signaling pathway is another crucial regulator of satellite cells because the specific depletion of RBPJ, the DNA binding factor essential for mediating canonical Notch signaling, was shown to induce spontaneous differentiation and loss of muscle satellite cells during quiescence, and subsequent injury [48]. The latest research find has shown that, in a mouse model with satellite cell—specific conditional knockout (cKO) of Tmem30, there was a decrease in Pax7+ and MYH3+ satellite cells, indicating reduced satellite cell proliferation and impaired myoblast proliferation in skeletal muscle regeneration [49].

As aging progresses, the function of satellite cells declines and there is impairment in the ability of muscle regenerate. Initially, decreasing the concentration of growth hormone in the body will increase myostatin expression, which inhibits satellite cell division [50]. Furthermore, changes in the microenvironment may cause decrease in the potential of satellite cells to proliferate. Researchers have found that IGF-1 levels in satellite cells decreased in old age, and impaired regeneration in aging muscle was shown to be associated with decreases in the number of Delta ligands in the Notch pathway. In addition, studies on elderly subjects have shown alterations in the Wnt pathway and the muscle progenitor cells are not differentiating into myofibers but fibroblasts or adipocytes [51].

## 4. Effects of Oxidative Stress on Skeletal Muscle

An inappropriate concentration of ROS can damage skeletal muscle and give rise to metabolic diseases like diabetes. Excessive ROS can trigger mitochondrial for degradation, including mitochondrial DNA damage, electron transport chain abnormalities and disruption of the membrane potential. However, in this section, we discuss damage to myogenesis and muscle regeneration caused by an inappropriate concentration of ROS.

### 4.1. Effects of Oxidative Stress on Myogenesis

Muscle cells consume a large amount of oxygen, which is significantly increased during exercise. This O_2_ consumption is associated with the continuous generation of ROS/RNS [52]. Initially, OS appears to mostly reduce the efficiency of myogenic differentiation, while a high concentration of ROS may contribute to the loss of myoblast function and increased myoblast cell death. ROS can lead to reduction in intracellular GSH pool, which causes NF-κB activation, resulting in reduced MyoD expression [53]. Besides, NF-κB mediated activation of YY1, a myogenic transcriptional inhibitor, may also be a aim of the ROS-mediated silencing of myogenic differentiation in myoblasts [54]. It was also found that activity of the p21 promoter declined in myoblasts in response to high ROS and that more pronounced apoptosis occurred in myoblasts than in myotubes [55]. Elevated ROS levels can also affect muscle contractile function by influencing the sensitivity of muscle fibers to Ca^2+^ [56]. However, ROS production is a stimulus for the adaptation of skeletal muscle to exercise training [57]. Exposure to hydrogen peroxide exposure has also been shown to augment the expression of key antioxidant enzymes in myotubes [32] and increase peroxisome proliferator activated receptor-γ coactivator-1 protein-α (PGC-1α) promoter activity and mRNA expression [58].

Taking the reported findings together, controversy remains about how the redox cell environment affects myogenesis, and researchers need to continue unearthing more convincing evidence.

### 4.2. Effects of Oxidative Stress on Muscle Regeneration

Muscle repair is dependent upon the function of satellite cells (SCs); SCs regenerate and differentiate into skeletal muscle after acute or chronic injury. Substantial evidence has indicated that excess ROS leads to impaired muscle regeneration, mainly by affecting skeletal muscle satellite cell function. For example, Tfr1 deletion in SCs results in the irreversible depletion of SCs and cell-autonomous defect in SCs proliferation and differentiation. This in turn lead impairment of skeletal muscle regeneration, followed by labile iron accumulation, lipogenesis, and decreased Gpx4 and Nrf2 protein levels, resulting in defects in ROS scavenging [59], and abrogation of Nrf2/antioxidant signaling promotes oxidative stress and impairs skeletal muscle regeneration as well [60]. Moreover, excessive ROS levels will cause muscle satellite cells senescence during muscle regeneration [61]. However, the production of ROS is not absolutely harmful in muscle regeneration. In a study on the tail regeneration using gecko model, ROS was found to be necessary for the regeneration [62], furthermore, ROS can help macrophages engulf cellular debris produced by the injury process, which can accelerate muscle regeneration [37].

### 4.3. Other Effects of Oxidative Stress

Besides its influence on myogenesis and muscle regeneration, OS influences cell differential (Figure 2). Adipose accumulation was observed in denervated muscle and atrophied muscle of the elderly. One of the characteristics of adipogenesis is the accumulation of ROS, and it induces the adipogenic differentiation of skeletal myofiber-associated cells [63] Myoblasts and brown adipocytes are known to share common Myf5+ progenitor cell and the level of bone morphogenetic protein 7 (BMP-7), a TGF-β family member can decided the cell fate. ROS overproduction can activate NF-κB signaling pathway, then upregulate S100B and accumulate that, ultimately leading to myoblast transition into brown adipocytes [64].

Furthermore, systemic oxidative stress is related to exercise intolerance and skeletal muscle abnormalities in patients with chronic heart failure. The most important change in such patients is abnormal intramuscular energy metabolism [65].

## 5. Oxidative Stress and Muscle Diseases

Numerous data suggests that OS plays a critical role in the pathophysiology of various muscle diseases, including dystrophy, atrophy, and hypertrophy, although no effective antioxidant therapy has been established. The balance between ROS production and antioxidant defense plays a decisive role in maintaining redox homeostasis in muscle. As we mentioned previously, the excessive production of ROS leads to oxidative stress and cell damage by affecting protein function, membrane structure, DNA integrity, and lipid metabolism, thus interfering with the homeostasis and function of skeletal muscle. Therefore, OS has been recognized as one of the most important causes of muscle injury in these diseases, besides the pathogenic consequences of individual gene mutations [23,66].

### 5.1. Oxidative Stress and Muscular Dystrophy

Muscular dystrophy (MD) is a group of genetic muscle diseases with heterogeneous phenotypes and genotypes. Its histopathological manifestations include inflammation, degeneration, necrosis, and fibrosis. These diseases are characterized by weakness, muscle atrophy, and progressive muscle degeneration, which can eventually lead to impaired activity and premature death. The most common type of MD is Duchenne muscular dystrophy (DMD), which is an incurable hereditary X-linked recessive muscle disease and the most serious progressive muscular dystrophy. DMD is caused by loss function mutation of the gene encoding dystrophin, a structural protein located on the cytoplasmic surface of muscle membrane [67]. DMD is a debilitating disease that causes progressive muscle atrophy and weakness. Several aberrant processes (e.g., intracellular calcium homeostasis inflammation and ROS metabolism) have been implicated as early events in the pathophysiology of this diseases [68].

OS may be an significant factor of DMD muscle pathology. The expression of most enzymes related to antioxidant defense was shown to be enhanced in DMD and *mdx* mice. *mdx* (X-linked muscular dystrophy) mice are the most commonly employed models in DMD research. Muscle biopsies from DMD patients show increased oxidative stress compared with that of controls, and increased NADPH oxidase (Nox2) activity as an early event in disease on set [69,70]. OS has also been considered as a mechanism behind muscle injury and myasthenia in humans and *mdx* mice [71,72]. A study has been shown that muscle fibers lacking dystrophin appear to be more vulnerable to OS [73]. Moreover, another study on the treatment of mdx mice with N-acetylcysteine (NAC), an effective antioxidant, and their improvement of muscular dystrophy, further supports the assertion that increased oxidative stress is the key pathological mechanism of DMD muscle degradation [74]. However, conditional expression of the Polycomb group protein Bmi1 in muscle satellite cells remarkably improved muscle function in *mdx* mice by upregulating metallothionein 1 (Mt1).. Upregulating Mt1 reduces muscle atrophy caused by oxidative stress, and Mt1 is closely associated with defense against oxidative stress induced cellular damage [75]. SOD1 expression was also significantly increased in dystrophin deficient muscles, and muscle specific overexpression of SOD1 was shown to enhance lipid peroxidation in muscles [75]. In a DMD mouse model, the level of the Bromodomain and extra-terminal domain (BET) protein BRD4 was found to be significantly increased, while the pharmacological inhibition of the BET protein was shown to limit oxidative stress and muscle damage [76].

Meanwhile, the delivery of catalytic mimetic of SOD and CAT (EUK-134), which eliminates both O_2_^•−^ and H_2_O_2_, to *mdx* mice reduced markers of OS in the diaphragm [77]. Oxidative damage in DMD is also due to a decrease in f GSH levels as a result of reduced activity of γ-glutamyl cysteine ligase, the limiting enzyme for GSH synthesis [77]. Additionally, greater activity of GSH-metabolizing enzymes, GPX and glutathione reductase, with concomitant increase in the GSSG: GSH ratio suggested acute oxidative stress in the hind limb muscles of *mdx* mice. ROS can also respond to oxidative stress via upregulating antioxidant enzymes, such as GPX and CAT [78].

In addition, The interaction between increased intracellular calcium levels and inflammation can exacerbate oxidative stress level. Increased intracellular calcium levels cause an increase in mitochondrial calcium concentration, which affects ATP synthesis. Increased cy. At the same time, increased ATP cause not only higher oxygen consumption but also increased electro flow, which increase the level of ROS [79,80]. In addition, membrane damage during the progression of DMD stimulates the degranulation of muscle—resident mast cells and activation of an immune cell cascade [81]. Immune cells, such as neutrophils and macrophages, additionally produce ROS to promote phagocytosis. Although the destruction of sarcolemma integrity is a common feature of different types of MD, the mechanisms of abnormal Ca^2+^ handling and increased OS that underlie the dystrophic phenotype are far from being comprehensively understood. It is currently believed that the subsequent pathogenic rise in myoplasmic Ca^2+^ leads to mitochondrial Ca^2+^ overload and uncontrolled ROS/RNS generation. Interestingly, the interaction between altered Ca^2+^ signaling and excessive mitochondrial ROS production has been proven to be a key pathological mechanism, not only in MD but also in other muscle diseases such as malignant hyperthermia and central core diseases [82]. However, it is unclear whether inhibiting Ca^2+^ entry can reduce oxidative stress.

In conclusion, oxidative stress may act upstream of the pro-inflammatory signal in *mdx* mice, and antioxidant therapy can be used to protect delay muscle atrophy in DMD. A method of preventing the pathology of dystrophin deficient mice by using different antioxidants has recently been intensively studied and reviewed [83,84,85] In addition, low-intensity endurance exercise has been shown to have a beneficial effect on the skeletal muscle of *mdx* mice. It reduced the protein carbonyls in mdx muscle and markers of oxidative stress [71]. A beneficial effect of regulating OS was also achieved by low-level laser therapy, which was found to reduce ROS and increase the activity of antioxidant enzymes including GPX, SOD and CAT [86,87]

### 5.2. Oxidative Stress and Muscular Atrophy

In theory, OS can lead to muscle atrophy by increasing proteolysis and/or inhibiting protein synthesis. Two of these pathways, the IGF1-Akt-Mtor pathway and the myostatin-Smad2/3 pathway have a very important role, with the former positively regulating and the latter negatively regulating [88]. Skeletal muscle mass increases during postnatal development through a process of hypertrophy. When there is an imbalance between protein synthesis and degradation, it can cause muscle hypertrophy or muscle atrophy. OS and autophagy are considered as the primary causes of skeletal muscle atrophy. Too much ROS can be activated by the forkhead box class O (FOXO) transcription factor in skeletal muscle [89].

Poor prognosis of diseases can lead to excessive loss of muscle mass, including cancer, heart failure, diabetes, etc. First, chronic kidney disease (CKD) can cause a variety of complications, including infection and muscle atrophy. Moreover, amyotrophic lateral sclerosis (ALS) is a rapidly progressive neuromuscular disease characterized by motor neuron death and devastating skeletal muscle wasting, which can also cause muscle atrophy. An excellent review covering the mitochondrial dysfunction, neuronal degeneration, and muscle atrophy in ALS has been published [90]. Furthermore, rheumatoid arthritis (RA) is a chronic inflammatory disease characterized by synovitis and the presence of serum autoantibodies, and numerous reviews have demonstrated ROS/RNS production in RA-induced muscle weakness [91]. Sarcopenia is another disease that causes muscle atrophy.

Skeletal muscle mass, function, and repair capacity all progressively decline with aging, and are associated with high levels of muscle ROS in vivo, which cause increased apoptosis and cell death and reduced myoblast differentiation, leading to poor muscle repair. Sarcopenia is a disease that results in loss of skeletal muscle mass and function with age, and leads to low levels of physical activity. This disease is exacerbated by obesity and metabolic disorders [92]. Severe muscle atrophy has also been identified in obese model mice harboring significant PMAT relative to respective control nonobese mice. Moreover mitochondrial regulation of muscle metabolism and mitochondrial dysfunction may play a role in sarcopenia [93].

To understand the molecular mechanisms involved, we also discuss muscle atrophy and OS as well. Firstly, altered redox signaling, which can affect regulatory transcriptional activators, increases the expression of genes encoding key components of the autophagy and proteasome systems [94]. Second, due to inactivity or disuse of skeletal muscle, the ROS can activate caplain and caspase-3 [94,95,96]. Finally, ROS accelerate the proteolysis of muscle fibers by oxidizing muscle proteins and enhancing their susceptibility to proteolysis [97]. ROS excessive production can affect muscle protein synthesis, which can lead to atrophy. In this regard, cellular protein synthesis occurs through a complex network of signaling pathways, which ultimately translates mRNA into specific proteins. The rate of protein synthesis is mainly controlled by translation efficiency. Studies have shown that ROS can block mRNA translation at the initial stage and inhibit protein synthesis [98,99,100,101]. For example, reduced phosphorylation of mammalian rapamycin substrate targets, such as eukaryotic initiation factor 4E binding protein and p70 S6 kinase [98]. The observation that muscular tube exposure to ROS (i.e., H_2_O_2_) leads to fiber atrophy provides a another evidence supporting the idea that oxidants promote muscle atrophy [94,96,102]. OS was shown to promote the expression and allosteric regulation of proteases of muscle atrophy. For instance, treatment of myotube to H_2_O_2_ has been shown to increase FoxO3a signaling and the expression of genes encoding important E3 ligases, which are involved in the ubiquitin-proteasome system of proteolysis, and autophagy components [96]. At the same time, OS induced by inactivity is a necessary condition for the increased expression of E3 ligase atrogin-1 and MurF-1 in skeletal muscle in vivo [103,104]. In addition, during prolonged periods of inactivity, it seems that oxidants are needed to activate calpain and caspase-3 in motor and respiratory muscles [103]. Besides to protease activation, ROS-induced protein oxidation provides the fourth piece of evidence linking OS to accelerated muscle protein breakdown. The prevention of OS caused by inactivity can delay or prevent wasting muscle atrophy. Many studies have shown that using different oxidant-scavenging compounds treating animals can prevent disuse muscle atrophy [103,104,105,106,107,108,109].

### 5.3. Oxidative Stress and Muscle Hypertrophy

The soleus and gastrocnemius muscles were ablated, which resulted in compensatory hypertrophy of the plantar muscle [110]. Two weeks after removing the soleus and gastrocnemius muscles, an increase in the muscle mass of the plantar muscles of about 40% was observed. This hypertrophy was associated with a significant increase in the content and activity of Silent Mating Type Information Regulation 2 Homolog 1 (SIRT1) (*p* < 0.001). SIRT1 regulates the levels of Akt, endothelial nitric oxide synthase and GLUT4. SIRT1 levels were found to be associated with muscle mass, Pax7,, nicotinamide phosphoribosyl transferase (Nampt) and proliferating cell nuclear antigen (PCNA) levels [110]. These findings indicate that the redox state of cells affects muscle growth, at least in this model. We also found that the increased protein levels of K63 and Muscle Ring Finger 2 (MuRF2) may also be important enhancers of muscle mass and reported that the levels of microRNA (miR)1 and miR133a decrease in hypertrophy and are negatively correlated with muscle mass, SIRT1 and Nampt levels. These results indicate a strong correlation between SIRT1 and hypertrophy induced by overload [110].

In the case of cardiac hypertrophy, the ubiquitin E3 ligase TRAF6 was found to be elevated in human and mouse hypertrophic hearts. In terms of the mechanism involved in this, it was found that TRAF6 is regulated by ROS generated during the progression of hypertrophy, and ROS trigger their own ubiquitination, leading to the formation of TRAF6-TAK1 interaction, which is essential for cardiac remodeling. Moreover, it has been reported that REGγ, an activator of the 11S proteasome, binds and activates the 20S proteasome to promote the degradation of several proteins [111]. Under pressure overload, an increase in REGγ is associated with the nuclear export of FOXO3a, which subsequently leads to a decrease in MnSOD accompanied by an increase in ROS [111]. In the same study, ectopic expression of MnSOD was shown to reduce REGγ-mediated ROS accumulation.

For a more intuitive understanding, we have drawn table to describe the relationship between OS and muscle diseases (Table 2). 

## 6. Antioxidant Therapy

Under different conditions, endogenous antioxidants cannot completely prevent oxidative damage there may also be a disturbance in the balance fo endogenous antioxidants and increase OS. It would thus be reasonable to attempt to establish treatments alleviate high levels of ROS and therefore limit protein degradation and apoptosis in skeletal muscle. Many studies have confirmed the need for scavenging reactive oxygen species. We describe some of the antioxidant treatments in this section.

### 6.1. Antioxidants

Because OS is caused by an imbalance between the production and removal of ROS, reducing ROS production and increasing antioxidant capacity are two significant approaches to antioxidant therapy. Among these approaches, the administration of antioxidants is the most widely used. Antioxidants are substances that effectively inhibit or delay the occurrence of oxidation reactions. The antioxidant protects the hazard on it by preventing its oxidative conversion via ROS or RNS. Primary antioxidants sacrificially act by breaking the chain reactions involved in lipid peroxidation through inhibiting of radical initiation, upon reacting with lipid radicals (L^•^), or by radical propagation upon reacting with lipid peroxyl (LOO^•^) or alkoxyl (LO^•^) radicals, as comprehensively reviewed elsewhere [112].

Among the available antioxidant treatments, ascorbic acid (vitamin C) and α-tocopherol (vitamin E) are the most commonly used. Ascorbic acid is hydrophilic antioxidant that accumulates in the aqueous phase of the cell, which can protect cell components against free radicals commonly formed during metabolic reactions. α-Tocopherol is known to regulate redox balance in the body due to its high concentration among the lipid soluble vitamin groups, and its ubiquity in the body, including in cell membranes and lipoproteins. Ascorbic acid has also been reported to have a positive therapeutic effect in cancer prevention, maintaining redox balance in the central nervous system, and exerting radioprotective effects [113]. Moreover, vitamin E protects cells against OS through inhibiting of lipid peroxidation [12].

Carnosine is a dipeptide that is presenting in high concentrations in mammalian skeletal muscle and is naturally occurring. It has been shown to affect muscle contraction, preventing the accumulation of by-products of oxidative metabolism and maintain muscle acid-base balance. The activity of carnosine differs between in myoblasts under OS and those under basal conditions. In oxidative stress conditions, carnosine pretreatment was found to increase the mRNA levels of the Nrf2 transcription factor and several of its downstream genes known to reduce H_2_O_2_. However, it was reported that supra-physiological carnosine concentrations are probably cytotoxic to myoblasts [114]. L-Carnosine and its isomer D-carnosine have been proven to have significant cytoprotective effects by reducing OS in different cell types, which is considered to occur via the following mechanisms: inhibition of hydrogen peroxide induced cytotoxicity, decreases of intracellular reactive oxygen, and nitrogen species, decrease of the intracellular concentration of superoxide anions (O_2_^−^) [115,116].

Creatine is composed of arginine, methionine and glycine. which can be synthesized via cooperation among various organs, such as the liver, pancreas, and kidneys [117], which can increase the activity of antioxidant enzymes and the ability to eliminate ROS and RNS. Other antioxidants, such as taurine, piceatannol (PIC) and enzymatically modified isoquercitrin (EMIQ) also play important roles in decreasing ROS level. Taurine is mainly used for muscle recovery after exercise, and its antioxidant effect has been demonstrated in both myotubes and muscle fibers. The latest research has shown that taurine can protect myoblasts against decreased viability due to cisplatin, promote cellular ROS clearance and preserve the expression of MyoD1, myogenin, and MHC as well as the differentiation ability of myotubes [118]. In addition PIC and EMIQ can rescue body weight decline in aged mice, while PIC can also increase locomotor activity and suppress carbonylated protein in the skeletal muscle of aged mice [119].

### 6.2. Chinese Herbs

In China, there is a long history of using herbs to treat numerous diseases. In recent years, researchers have found that some herbs may cure muscle atrophy via antioxidant activity. For example, curcumin was found to exert antioxidant and anti-inflammatory effects on C2C12 myoblast cells. It increased DPPH radical scavenging activity in a dose-dependent manner, moreover enhancing the levels of hemeoxygenase 1 (HO-1), NAD(P)H-dependent quinine oxidoreductase (NQO)-1, glutamate cysteine ligase catalytic subunit (GCLC) [120]. Meanwhile, curcumin has been confirmed to alleviate chronic kidney disease-induced muscle atrophy by inhibiting glycogen synthase kinase (GSK)-3β [121].

Moreover, atractylenolide III is the main active component of *Atractylodes* rhizome, which has been reported to have antioxidant effects. It can alleviate mitochondrial damage and increase the activity of antioxidant enzymes, thus reducing the production of ROS. Moreover, atractylpenolide III decreased the expression levels of the inflammatory factors interleukin-1 β and tumor necrosis factor-α and inhibited the levels of the prooxidant markers, while increasing the expression levels of the antioxidant enzymes catalase, superoxidant dismutase and glutathione peroxidase [122,123].

Silymarin is another useful antioxidants from natural. The antioxidant actions of silymarin have been reviewed by Surai [124]. The mechanism of silymarin antioxidant can be divided into five parts. 1 preventing free radical formation by inhibition of ROS-producing enzymes; 2 scavenging of free radicals; 3chelation ions in intestine; 4 promotes the synthesis of protective moleculesto provide protection against stressful stimuli [124]. 5 activation of antioxidant enzymes [125,126]. There are other herbs with antioxidant effects, but they are less studies, such as lotus plumule [127], radix astragali [128].

In order to better understand the mechanism of action on different antioxidants, we have summarized the table for detailed explanation (Table 3). 

## 7. Conclusions and Perspectives

This review describes the effects of ROS in skeletal muscle on myogenesis and muscle regeneration. It summarizes some of the pathogenic mechanisms of ROS in muscle diseases, including dystrophy, atrophy, and muscle hypertrophy. It also lists effective drugs currently used for treating OS-induced muscle diseases, such as antioxidants and herbal medicines.

Under physiological conditions, the normal maintenance of cellular metabolism and other functions is linked to the balance of ROS and RNS. At the same time, the redox system regulates the function of multiple signaling proteins that affect cell survival in different ways, such as signal transduction and transcriptional regulation [2]. Oxidative stress occurs as a result of the excessive production of ROS in the body, as well as impaired antioxidant capacity.

When the production of reactive oxygen species is too high, it can lead to a range of diseases such as type II diabetes, cancer, Alzheimer’s disease, as well as promoting muscle aging [131].

Some limitations of this study should be addressed. First, the etiology of muscle-related disorders involves other factors besides a high level of ROS. For instance, prolonged inactivity, inadequate nutritional intake and insufficient protein supply can also lead to muscle atrophy. Second, we do not describe ROS and aging in detail. Increased ROS activity has been implicated in the processes underlying aging in all species. Skeletal muscle of aging organisms exhibits increased expression of cytoprotective proteins through the activation of redox-sensitive transcription factors, resulting in a severely diminished ability to respond to increased ROS production. Many reviews have concluded that age-related changes in skeletal muscle occur due to ROS generation and adaptive responses to ROS [132,133]. Third, most of the surveyed studies on muscle diseases were conducted on mice, rats or rabbits, and not on humans, so the mechanisms and human medicines are somewhat limited. Initially, it was found in rats that xanthine oxidase produces superoxide in the cytosol of contracting skeletal muscles [134]; however, compared with that in rats, human skeletal muscle contain lower levels of xanthine oxidase, and the question of whether xanthine oxidase plays an important role in superoxide production in human skeletal muscle remains controversial [135,136]. In addition, a recent study showed that experimental mouse models do not accurately represent the human response to inflammatory diseases [137], so we need to find more appropriate animal models for studying of inflammation associated with ROS. Fourth, in addition to antioxidants and traditional Chinese herbs, a growing number of novel drugs have emerged for the effective treatment of diseases associated with oxidative stress, including melatonin [138], ionic silicon [139], and selenium [140].

It is important to detect and prevent muscle-related diseases using currently available tools including genetic testing, diagnostic algorithms, muscle biopsies, EMG, and other conventional diagnostic tests [141]. We are confident that new experimental studies and comprehensive reviews on these crucial themes will be performed in the future.

## Figures and Tables

**Figure 1 antioxidants-11-00755-f001:**
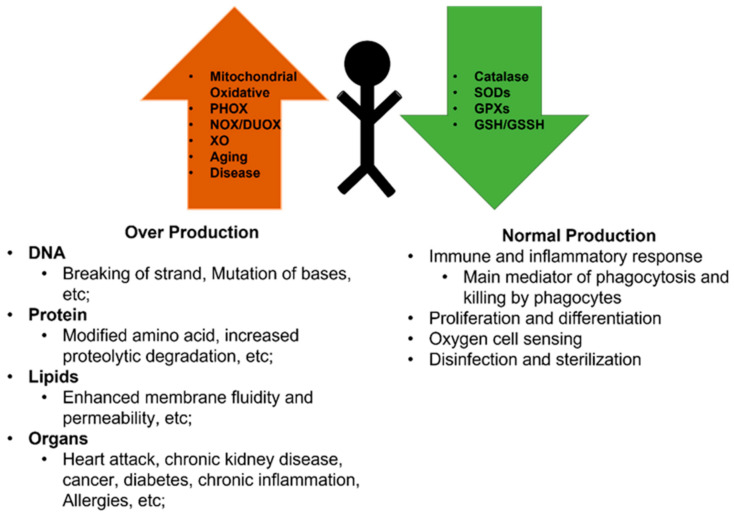
Schematic representing the reactive oxygen species and reactive nitrogen species influence in different conditions. The red arrow represents the causes of ROS and RNS generation, the green arrow represents the cause of elimination.

**Figure 2 antioxidants-11-00755-f002:**
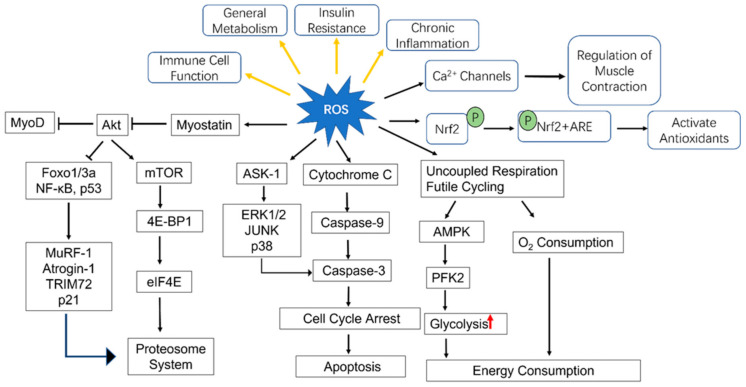
Effects of ROS on myogenic processes and other aspects. NF-κB: nuclear factor κB, MuRF-1: muscle RING-finger protein-1, TRIM72: tripartite motif 72, 4E-BP1: eukaryotic initiation factor 4E-binding protein, eIF4E: eukaryotic initiation factor (eIF) 4E, ASK-1: apoptosis signal-regulating kinase 1, ARE: antioxidant response element; Nrf2: nuclear factor erythroid 2-related factor 2, AMPK: Adenosine 5′-monophosphate (AMP)-activated protein kinase, PFK2: phosphofructokinase 2, mTOR: mechanistic target of rapamycin, ERK1/2: extracellular signal-regulated kinases 1/2.

**Table 1 antioxidants-11-00755-t001:** Common reactive oxygen species and reactive nitrogen species in the body.

Type of ROS and RNS	In Vivo Concentration (M)	Major Generating Reaction
Superoxide anion (O_2_^−^)	10^−10^	NADPH + 2O_2_ ⇄ NADP^+^ +2O_2_^−^ + H^+^ 2O_2_^−^ + H^+^ → O_2_ + H_2_O_2_
Hydroxyl radical (HO^•^)	10^−15^	Fe^2+^ +H_2_O → Fe^3 +^ +OH^−^ +·OH
Hydrogen peroxide (H_2_O_2_)	10^−7^	O_2_+ H_2_O + Hypoxanthine ⇄ H_2_O_2_ + Xanthine O_2_+ H_2_O + Xanthine ⇄ H_2_O_2_ + uric acid
Alkoxy radical (RO^•^)		RO2· + NO → RO· + NO_2_
Nitric oxide (NO^•^)		2 L-arginine + 3NADPH + 3H ^+^ +4O ⇄ 2NO· + 4 H_2_O + 3NADP^+^
Peroxynitirite (ONOO^−^)		NO·+ O_2_^−^ → ONOO^−^

The table was modified from Refs. [12,13,14].

**Table 2 antioxidants-11-00755-t002:** Oxidative Stress and Muscle Diseases.

Muscle Diseases	Causes	Results	Reference
muscular dystrophy	increased Nox2		[69,70]
muscular dystrophy	increased SOD1	enhanced lipid peroxidation in muscle	[75]
muscular dystrophy	upregulation Mt1	delay muscle wasting through reduction of ROS-induced oxidative stress	[75]
muscular atrophy	high level of ROS	mitochondrial dysfunction, activate FOXO transcription factor	[89]
muscular atrophy	high level of ROS	accelerate proteolysis in muscle fiber, increasing sensitivity to proteolysis, hinder muscle protein synthesis	[97]
muscle hypertrophy	increases sirt1	the levels of Akt, endothelial nitric oxide synthase and GLUT4, PAX7, PCNA, NAMPT	[110]
muscle hypertrophy	increase ubiquitin E3 ligase TRAF6	TRAF6-TAK1 interraction	[111]
muscle hypertrophy	increase REGg	nuclear output of FOXO3, decrease MnSOD, increase ROS	[111]

**Table 3 antioxidants-11-00755-t003:** Common Antioxidants.

Antioxidants	Mechanism	Reference
Vitamin C	Protect cell components against free radicals, protect mitochondrial,	[113,129]
Vitamin E	Inhibit of lipid peroxidation	[12]
Carnosine	Inhibition of hydrogen peroxide induced cytotoxicitym decrease of intracellular concentration of superoxide anions	[115,116]
Creatine	Increase the activity of antioxidant enzymes, eliminate ROS and RNS	[117]
Taurine	Protect myoblasts against decreased viability due to cisplatin, promote cellular ROS clearance, preserve the expression of MyoD1, myogenin, and MHC	[118]
Piceatannol (PIC) and Enzymatically modified isoquercitrin (EMIQ)	Rescue bodyweight decline in aged mice, increase locomotor activity, suppress carbonylated protein	[119]
Curcumin	Increase DPPH radical scavenging activity, enhance the level of HO-1, GCLCand NQO-1 Inhibit glycogen synthase kinase (GSK)-3β	[120,121,130]
Atractylenolide III	Alleviate mitochondrial damage, downregulate the expression level of IFN-1β, TNF-α, enhance the expression levels of antioxidant enzymes catalase, superoxidant, dismutase, glutathione peroxidase.	[122,123]
Silymarin	Prevention of free radical formation, scavenging of free radicals action, chelation of ions in intestine, promote the synthesis of protective molecules, activate antioxidant enzymes	[124,125,126]

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
