# Peer review of "The Role of Oxidative Stress in Skeletal Muscle Myogenesis and Muscle Disease"

_antioxidants, 2022, doi:10.3390/antiox11040755_

Round 1

Reviewer 1 Report

Title:  The role of oxidative stress in skeletal muscle myogenesis and muscle disease

Journal: Antioxidants

Date: 03-15-2022

Recommendation

Accept after Minor revision

Comments

The authors reviewed about "the role of oxidative stress on skeletal muscle in physiological and pathological conditions." Also, antioxidant therapy for treatment of muscle diseases. Overall, the manuscript was well-organized and well-written. However, there are some weaknesses should be improved. Please see the follows as.

Major Points

  1. A schematic figure showing how the ROS and RNS regulates myogenesis and muscle regeneration regarding '4. Effects of Oxidative Stress on Skeletal Muscle" is needed for helping readers to figure out it easily. Please include the various transcription factors such as MRFs.

  1. Line 126-127: "For example, exposure to hydrogen peroxide has been shown 126 to augment the expression of key antioxidant enzymes in myotubes [20]" Please include the details what key antioxidant enzymes are.  

  1. Line 153-155: "After muscle injury, muscle satellite cells are stimulated by the various signals arising from the damaged environment, and can migrate toward the site of injury and re-enter the cell cycle to undergo proliferation." Which various signals are involved in? The authors should include the details explanation.

  1. Line 450-465: There are so many natural products including Chinese herbs are known for exerting anti-oxidant activities. However, very limited numbers of references were introduced. More references and contents should be added.

  1. A summarized table for '6. Antioxidant Therapy' is recommended for help readers to find out their interesting parts for therapeutical applications.

Minor points

  1. Line 33: reactive oxygen (ROS) -> reactive oxygen species (ROS); It should be edited.

  1. Line 34: nitrogen species (RNS) -> reactive nitrogen species (RNS); It should be edited.

  1. Line 126-130 and line 194-197: The very similar sentences were repeated. Please check and revise them.

Author Response

1. Schematic figure showing how the ROS and RNS regulates myogenesis and muscle regeneration regarding '4. Effects of Oxidative Stress on Skeletal Muscle" is needed for helping readers to figure out it easily. Please include the various transcription factors such as MRFs.

Response: Thanks for the meaningful suggestions. We have drawn a figure containing the effecting of elevated ROS on different aspects, including proteosome system , cell apoptosis, myogenesis, muscle contraction and energy consumption.

2. Line 126-127: "For example, exposure to hydrogen peroxide has been shown 126 to augment the expression of key antioxidant enzymes in myotubes [20]" Please include the details what key antioxidant enzymes are.  

Response: We are very grateful for your constructive suggestions. And we added three antioxidant enzymes in this paragraph: CuZnSOD, MnSOD and GPx.

3. Line 153-155: "After muscle injury, muscle satellite cells are stimulated by the various signals arising from the damaged environment, and can migrate toward the site of injury and re-enter the cell cycle to undergo proliferation." Which various signals are involved in? The authors should include the details explanation.

Response: Thank you for your meaningful suggestions. We added some pathways in satellite cell activation and migration such as Wnt/PCP. Furthermore, we added satellite cell proliferation pathways including MEK/ERK, JAK/STAT1/STAT3 and PI3K/Akt.

4. Line 450-465: There are so many natural products including Chinese herbs are known for exerting anti-oxidant activities. However, very limited numbers of references were introduced. More references and contents should be added.

Response: Thank you for your constructive comments. According to the reviewer’ suggestions, we have added three herbs related to antioxidant functions, including Silymarin, lotus plumule and radix astragali.

5. A summarized table for '6. Antioxidant Therapy' is recommended for help readers to find out their interesting parts for therapeutical applications.

Response: Thank you for your suggestion. Based on your suggestions, we have added a table in part 6, including antioxidant drugs and herbal medicines.

1. Line 33: reactive oxygen (ROS) -> reactive oxygen species (ROS); It should be edited.

Response: Thank you again and we have made a change to the error.

2. Line 34: nitrogen species (RNS) -> reactive nitrogen species (RNS); It should be edited.

Response: Thank you again and we have made a change to the error.

3. Line 126-130 and line 194-197: The very similar sentences were repeated. Please check and revise them.

Response: Thank you for your suggestions, and we deleted the sentences at the part 2, line 126-130.

Reviewer 2 Report

Comments and Suggestions for Authors

The authors in the manuscript „The role of oxidative stress in skeletal muscle myogenesis and muscle disease“ have tried to examine the effect of oxidative stress on myogenesis, the relationship between oxidative stress and muscle homeostasis and physiopathology and provide targets for antioxidant therapy and repair.

The topic is interesting, but after careful revision of the manuscript I noted several points which the authors should address when if invited to prepare a revision:

  1. Table 1 – correct the mistakes (see Giorgio, M., Trinei, M., Migliaccio, E., and Pelicci, P. G. (2007) Hydrogen per-oxide: a metabolic by-product or a common mediator of ageing signals? Nat. Rev. Mol. Cell Biol. 8, 722–728).
  2. Paragraph 2.2.1 – there is no data presented about mitochondrial enzymes or complexes responsible for ROS generation in skeletal muscles. In my opinion, this paragraph is addressed to explain the effects of ROS on skeletal muscle.
  3. Paragraph 2.3 – lack of new information about ROS effect in muscles – which intracellular signal transduction pathways are involved in and how changes in ROS concentration affect them.
  4. Paragraphs 4.1 and 4.2 lack of consistency and novelty.
  5. 6.2 - please expand this paragraph and add new data about the effect of biologically active compounds of herbs on oxidative stress in muscle cells.

Author Response

1. Table 1 – correct the mistakes (see Giorgio, M., Trinei, M., Migliaccio, E., and Pelicci, P. G. (2007) Hydrogen per-oxide: a metabolic by-product or a common mediator of ageing signals? Nat. Rev. Mol. Cell Biol. 8, 722–728).

Response: Thank you for your careful reminder. Based on the literature you provided, we have read it carefully and modified the table 1, the concentration of superoxide anion in vivo is 10-10.

2. Paragraph 2.2.1 – there is no data presented about mitochondrial enzymes or complexes responsible for ROS generation in skeletal muscles. In my opinion, this paragraph is addressed to explain the effects of ROS on skeletal muscle.

Response: Thanks for your constructive comments. According to the reviewer' suggestions, we have supplemented data on mitochondrial enzymes or complexes responsible for ROS production in skeletal muscle.

3. Paragraph 2.3 – lack of new information about ROS effect in muscles – which intracellular signal transduction pathways are involved in and how changes in ROS concentration affect them.

Response: We are very grateful for your constructive suggestions. We have updated our incomplete statement. We have added relevant information about ROS in muscles, including the effect of ROS concentration or level on skeletal muscle, AMPK and NF-κB signal pathway, and PGC-1α/β-dependent antioxidant defense mechanisms.

4. Paragraphs 4.1 and 4.2 lack of consistency and novelty.

Response: Thanks for the meaningful suggestions. We have added some contents including NF-κB signaling pathway and the benefit about the ROS during muscle regeneration. At the part 4, we also added a graph to help understand the influence of ROS on different aspects.

5. 6.2 - please expand this paragraph and add new data about the effect of biologically active compounds of herbs on oxidative stress in muscle cells.

Response: Thanks for the meaningful suggestions. According to the reviewer’ suggestions, we have added three herbs related to antioxidant functions, including Silymarin, lotus plumule and radix astragali.

Round 2

Reviewer 2 Report

Accept